# The rs1883832 Polymorphism (CD40-1C>T) Affects the Intensity of IgA Responses after BNT162b2 Vaccination

**DOI:** 10.3390/ijms232214056

**Published:** 2022-11-14

**Authors:** Matthaios Speletas, Evangelos Bakaros, Athanasia-Marina Peristeri, Ioanna Voulgaridi, Styliani Sarrou, Vassiliki Paliatsa, Asimina Nasika, Maria Tseroni, Lemonia Anagnostopoulos, Kalliopi Theodoridou, Fani Kalala, Aikaterini Theodoridou, Barbara A. Mouchtouri, Sotirios Tsiodras, Hermann Eibel, Christos Hadjichristodoulou

**Affiliations:** 1Department of Immunology & Histocompatibility, Faculty of Medicine, University of Thessaly, 41500 Larissa, Greece; 2Laboratory of Hygiene and Epidemiology, Faculty of Medicine, University of Thessaly, 41222 Larissa, Greece; 3National Public Health Organization, 15123 Athens, Greece; 4Department of Microbiology, Andreas Sygros Hospital, National and Kapodistrian University of Athens, 16121 Athens, Greece; 5Fourth Department of Internal Medicine, School of Medicine, Attikon University Hospital, National and Kapodistrian University of Athens, 12462 Athens, Greece; 6Department of Rheumatology and Clinical Immunology, Medical Center and Faculty of Medicine, University of Freiburg, 79106 Freiburg, Germany; 7Center for Chronic Immunodeficiency, Medical Center and Faculty of Medicine, University of Freiburg, 79106 Freiburg, Germany

**Keywords:** CD40, rs1883832, IgA responses, COVID-19 vaccination

## Abstract

The effectiveness of coronavirus disease 2019 (COVID-19) vaccination strategies is affected by several factors, including the genetic background of the host. In our study, we evaluated the contribution of the functional polymorphism rs1883832 affecting the Kozak sequence of the *TNFSF5* gene (c.-1C>T), encoding CD40, to humoral immune responses after vaccination with the spike protein of SARS-CoV-2. The rs1883832 polymorphism was analyzed by PCR-RFLP in 476 individuals (male/female: 216/260, median age: 55.0 years, range: 20–105) of whom 342 received the BNT162b2 mRNA vaccine and 134 received the adenovirus-based vector vaccines (67 on ChAdOx1-nCoV-19 vaccine, 67 on Ad.26.COV2.S vaccine). The IgG and IgA responses were evaluated with chemiluminescent microparticle and ELISA assays on days 21, 42, and 90 after the first dose. The T allele of the rs1883832 polymorphism (allele frequency: 32.8%) was significantly associated with lower IgA levels and represented, as revealed by multivariable analysis, an independent risk factor for reduced anti-spike protein IgA levels on days 42 and 90 following BNT162b2 mRNA vaccination. Similar to serum anti-spike IgA levels, a trend of lower anti-spike IgA concentrations in saliva was found in individuals with the T allele of rs1883832. Finally, the intensity of IgA and IgG responses on day 42 significantly affected the prevalence of COVID-19 after vaccination. The rs1883832 polymorphism may be used as a molecular predictor of the intensity of anti-spike IgA responses after BNT162b2 mRNA vaccination.

## 1. Introduction

The COVID-19 pandemic represents one of the greatest challenges to public health during the 21st century [1]. The rapid development of COVID-19 specific vaccination strategies based upon newly implemented platforms—such as mRNA and adenovirus-based vector vaccines—resulted in a significant reduction of COVID-19-related morbidity and mortality [2]. However, similar to other vaccination strategies, the effectiveness of COVID-19 vaccines is affected by several factors, including host factors (such as age, sex, genetics, and comorbidities) and/or extrinsic factors (such as preexisting immunity, microbiota, and current treatments) [3].

Among host factors which may affect the effectiveness of vaccination strategies, in recent years both an increased interest and supporting evidence exists regarding the role of genetics, which is most evident from poor vaccination responses in patients suffering from primary immunodeficiencies [3,4,5]. In a more general way, Poland et al. also applied the term “vaccinomics” which encompasses the fields of immunogenetics and immunogenomics to understand mechanisms of heterogeneity in immune responses to vaccines [6]. Thus, single nucleotide polymorphism (SNP) in genes coding for cytokines or cytokine viral or vitamin receptors have been associated with variations in vaccine responses [3,7]. For example, the effectiveness of anti-measles vaccination has been associated with TLR3-6 functional polymorphisms [8], the persistence of IgG responses to MenC vaccines has been associated with polymorphisms in the TLR3 and CD44 genes [9], the adaptive cytokine responses to rubella vaccination are affected by polymorphisms in the TLR, vitamin A, vitamin D receptor, and by innate immunity genes [10], and the antibody responses after vaccination with HBsAg and hepatitis A are influenced by a functional polymorphism in the promoter of IL10 gene [11].

Among the genes that play a crucial role in adaptive immunity and may affect the effectiveness of vaccination strategies is *TNFRSF5*, encoding the tumor necrosis factor receptor member CD40. Activated by CD40L (CD154) expressed on the surface of T-helper cells, CD40 induces B cell proliferation in germinal center reactions and the differentiation of antigen-activated B cells into memory B cells and plasma cells. In addition, CD40 also regulates the development of immature dendritic cells [12,13]. A functional SNP located at -1 position from the start codon within the Kozak sequence (c.-1C>T, rs1883832) affects the initiation of the protein translation process as the T allele decreases the translational efficiency of *TNFRSF5* mRNA transcripts, resulting in approximately 15–30% less CD40 protein levels [14,15]. Previous studies have reported an association of this functional polymorphism with the susceptibility to several immune-related disorders, including Graves’ disease, multiple sclerosis, lymphomas, and sepsis [13,15,16,17,18], as well as with lower IgG levels after infection with Plasmodium vivax [19]. However, the role of rs1883832 polymorphism to humoral responses after COVID-19 vaccination is completely obscure.

Analyzing a large spectrum of immune gene polymorphisms to the intensity of antibody responses after COVID-19 vaccination, we initially observed a possible correlation between rs1883832 polymorphism and humoral responses after the BNT162b2 mRNA vaccine. We therefore evaluated the association of the rs1883832 polymorphism to anti-spike IgG and IgA immune responses at different time points after the anti-SARS-CoV-2 vaccination with mRNA- and adenovirus vector-based vaccines. According to our results, we consider that the rs1883832 polymorphism may be used as a molecular predictor of the intensity of IgA responses after the BNT162b2 mRNA COVID-19 vaccination.

## 2. Results

### 2.1. COVID-19 and Humoral Responses after Vaccination

As presented in Appendix A, immunization against SARS-CoV-2 with mRNA- and vector-based vaccines was well-tolerated. Local adverse effects were more common following the BNT162b2 mRNA-vaccine, while systemic adverse effects were most common following adenovirus-based vector vaccines. A total of 52 participants (10.9%) had a history of COVID-19 prior to vaccination, while 34 individuals (7.1%) were infected by SARS-CoV-2 after vaccination, including two participants who had been infected before vaccination. In all cases of COVID-19 following vaccination, the disease was either asymptomatic or mild. As presented in Appendix A, the great majority of enrolled participants displayed adequate spike-specific IgG humoral responses after vaccination independent of the type of vaccination platform. Interestingly and as already described by us in a recent report [20], IgA responses significantly waned in most participants three months after vaccination. In contrast to the BTN162b2 mRNA vaccinated group, most participants vaccinated with adenovirus-based vectors displayed anti-spike IgA levels below the cut-off of 10 U/mL at all time points of the study (Appendix A). 

### 2.2. Distribution of rs1883832 Polymorphism Frequency in the Vaccinated Participants of the Study

A total of 216 individuals carried the wild-type (CC) alleles, 208 were heterozygotes (CT), and 52 were homozygotes (TT) for the rs1883832 polymorphism (allele frequency 32.8%). In the group of individuals who received the BNT162b2 vaccine, 161 carried the wild-type alleles, 143 were heterozygotes, and 38 were homozygotes (allele frequency 32.1%), while in the group of participants receiving adenovirus-based vector vaccines, 55 carried the wild-type alleles, 65 were heterozygotes, and 14 were homozygotes (allele frequency 34.7%). The allele frequencies of rs1883832 polymorphism in the whole population and the subgroups of individuals receiving either the mRNA vaccination or the adenovirus-based vector vaccines were in Hardy–Weinberg equilibrium (*p* > 0.05, in all cases). This was similar with its allele frequency in other studies derived from the same geographic area around the Mediterranean Sea [21,22,23].

### 2.3. Correlations of rs1883832 Polymorphism with Humoral Responses after Vaccination

As presented in detail in Table 1 and Table 2 as well as in Figure 1 and Appendix A, the presence of the T allele was significantly associated with lower anti-spike IgA levels, determined 42 and 90 days after the first vaccination of BNT162b2, while anti-spike IgG levels remained unaffected; conversely, the adenovirus vector-based vaccines elicited adequate IgG responses but did not induce strong IgA responses, and the differences between the carriers of CC, CT, and TT alleles were not significant. However, when comparing carriers of the T allele in both the heterozygous and homozygous states, specifically as an entire group with participants carrying the wild-type alleles, we observed that IgA levels are significantly affected by the presence of rs1883832 polymorphism and for participants receiving the ChAdOx1 nCoV-19 vaccine on day 42 after vaccination (Appendix A). As expected, a similar association was observed in the participants receiving the BNT162b2 vaccine on days 42 and 90 (Appendix A) where IgA responses were stronger compared with those after adenovirus-based vector vaccines.

We recently reported that age and a history of COVID-19 (either before or after vaccination) were the most important risk factors significantly affecting the intensity and dynamics of IgG responses after vaccination; however, we observed age was not a significant factor for IgA responses, and only COVID-19 appeared to significantly influence IgA levels following vaccination [20]. Thus, we performed a further multivariable analysis in our cohort to elucidate the contribution of the rs1883832 polymorphism in the intensity of IgA levels after vaccination with the SARS-CoV-2 spike protein. As presented in detail in Table 3, the presence of the T allele (either in heterozygous or homozygous state) of rs1883832 polymorphism is an independent risk factor for significantly lower IgA levels on days 42 and 90 following BNT162b2 vaccination. In this context we also observed that in addition to the rs1883832 polymorphism, IgA levels on day 42 after vaccination are significantly affected by vaccination type, age, and history of COVID-19 prior to vaccination, while three months following vaccination, IgA levels are significantly affected by the type of vaccination and the exposure to SARS-CoV-2 prior or after vaccination (Table 3). On the other hand, the rs1883832 polymorphism did not affect the IgG levels; the latter on day 42 after vaccination are significantly affected by vaccination type, age, and history of COVID-19 prior to vaccination, while three months following vaccination, IgG levels are significantly affected by the type of vaccination and the exposure to SARS-CoV-2 prior or after vaccination (Appendix A).

Furthermore, following a booster vaccination, individuals with wild-type (CC) alleles displayed higher anti-spike IgA levels in saliva compared with those carrying the T allele in either the heterozygous or homozygous state. However, possibly due to the small number of participants analyzed, the difference was not marginally reached to be significant (Appendix A). Finally, comparing the expression levels of CD40 on the surface of B cells between carriers of the different alleles, we also observed lower MFI for CD40 in carriers of the T allele; however, even in this case, the difference was not significant (Appendix A).

### 2.4. Correlation of Humoral Responses and rs1883832 Polymorphism with COVID-19 Following Vaccination

As mentioned above, 34 participants (7.1%) were infected by SARS-CoV-2 after vaccination, including two participants who had also been infected prior to vaccination. Thus, we explored whether the intensity of antibody responses and the rs1883832 polymorphism affected the prevalence of COVID-19 after vaccination. As presented in Figure 2, we demonstrated the intensities of both IgA and IgG responses at day 42 after the first dose significantly affected the prevalence of COVID-19 after vaccination since the disease was more common in individuals with lower IgA and IgG levels. In this context and considering that as presented above, the IgA levels are significantly affected by the status of the rs1883832 polymorphism, we observed a similar trend for carriers of the T allele; however, the difference was not significant (allele frequencies 38.2% vs. 32.3% in participants with and without disease, respectively; *p* = 0.220).

## 3. Discussion

We clearly demonstrate a significant association of the rs1883832 polymorphism of *TNFRSF5* gene to anti-spike IgA antibody responses after vaccination with the SARS-CoV-2 S1 protein. This suggests that vaccination with BNT162b2 may be more effective against infection and, consequently, more protective against COVID-19 in CC allele carriers than in CT and TT allele carriers. This hypothesis is supported by the observation that individuals receiving adenovirus-based vector vaccines had lower anti-spike IgA levels, which correlated with a higher prevalence of COVID-19 after vaccination compared to those individuals who received the BNT162b2 vaccine (18.7% vs. 2.6%, Appendix A).

IgA plays a fundamental role in the immune function of mucous membranes, representing a first-line barrier to pathogens and other irritants. It is primarily produced in mucosal tissues; however, the bone marrow also provides an additional source, especially in humans [24,25]. Much of the produced IgA is transported into the secretory lumen, but some IgA naturally enters the lymphatic circulation and is transferred to blood via the thoracic duct [24]. IgA has a shorter half-life than IgG, which is estimated to be 4–6 days (compared to 25 days of IgG) [25]. Clearly, this fact justifies our findings, namely the more rapid decline of anti-SARS-CoV-2 IgA levels in blood compared to IgG levels and the absence of significant correlations on day 90 after the first vaccination.

Moreover, the protective immunity of IgA against respiratory viruses, such as respiratory syncytial virus (RSV) and influenza viruses, is already known [26]. A similar protective function of IgA has also been reported for SARS-CoV-2 [27,28], while low total serum IgA levels have been associated with a higher risk of SARS-CoV-2 infection and severe COVID-19 [29]. In this context, we and others have recently reported that individuals displaying lower anti-SARS-CoV-2 IgG and IgA levels exhibited a higher incidence of COVID-19 after vaccination [30,31,32].

In our study, we demonstrated a significant association between the rs1883832 polymorphism and IgA levels mainly after the BNT162b2 mRNA vaccination (Table 1 and Table 3, Figure 1). However, there is also a similar trend for participants receiving adenovirus-based vector vaccines, but the great majority of this latter group displayed lower (and below the cut-off of positivity) IgA levels compared to those observed after BNT162b2 vaccination (Table 1). Therefore, the intensity of IgA responses may be of particular importance for the prevention of COVID-19 after vaccination. This observation remains to be confirmed in further prospective studies.

In contrast to the correlation with anti-spike IgA levels, we did not find any association of rs1883832 polymorphism with anti-SARS-CoV-2 IgG levels after BNT162b2 vaccination. CD40 encoded by *TNFRSF5* gene is a key receptor regulating immunoglobulin class switch recombination (CSR) and differentiation into memory B cells and plasma cells; individuals with loss-of-function mutations in the *TNFRSF5* gene develop hyper-IgM syndrome type 3 (HIGM3), a severe primary immunodeficiency characterized by normal or elevated serum IgM levels and decreased levels or absence of both IgG and IgA, resulting in susceptibility to bacterial infections [33]. Interestingly, in vitro activation of B cells from HIGM3 patients results in the development of plasmablasts secreting low IgG levels, whereas IgA secretion was undetected [33,34], suggesting that CD40 is rather indispensable for IgA CSR and its secretion. Thus, the reduced CD40 surface expression linked to the T allele of the rs1883832 polymorphism could be responsible for lower IgA levels after vaccination.

We also acknowledge some limitations of our study. In particular, we cannot exclude the possibility that additional genetic or epigenetic factors affect the intensity of humoral responses after COVID-19 vaccination; a more comprehensive genetic approach (i.e., genome-wide association studies) may provide additional information. Moreover, IgA anti-SARS-CoV-2 antibodies in saliva were measured after a booster vaccination and not at the time points as initially planned, namely when we established the possible contribution of the rs1883832 polymorphism in IgA serum levels. However, according to data presented herein, we consider the rs1883832 polymorphism is an important factor affecting the intensity of IgA responses after BNT162b2 mRNA vaccination. It remains to be determined whether the rs1883832 polymorphism also affects both the severity and outcome of COVID-19 or the intensity of antibody responses after other types of vaccinations (i.e., attenuated or recombinant vaccines).

In conclusion, we suggest the functional rs1883832 polymorphism of the *TNFRSF5* gene may be a useful genetic marker to predict the intensity of IgA responses after COVID-19 mRNA vaccination. Further studies will uncover any associations between rs1883832 polymorphism and possible protection against SARS-CoV-2 infection and/or progression to severe COVID-19 in both vaccinated and unvaccinated individuals.

## 4. Materials and Methods

### 4.1. Study Population

A total of 342 individuals (male/female: 136/206, median age: 57.0 years, range: 27–105) who received the BNT162b2 mRNA vaccine (Pfizer-BioNTech) with an interval of 21 days between doses were initially enrolled in the study. Among them, 36 individuals (10.5%) displayed a positive medical history of COVID-19 (defined by a positive polymerase chain reaction result on a nasopharyngeal swab) at least three months prior to vaccination. Moreover, anti-nucleocapsid (anti-N) IgG anti-SARS-CoV-2 antibodies which are indicative of prior infection were additionally detected in four individuals at enrollment, suggesting a previous asymptomatic infection. Thus, the total number of study participants with a previous history of COVID-19 was 40 (11.7%). Furthermore, we enrolled 134 additional individuals (male/female: 54/80, median age: 49.0 years, range: 20–84) who received adenovirus-based vector vaccines. This included 67 individuals (male/female: 28/39, median age: 64.0 years, range: 33–84) who received the ChAdOx1 nCoV-19 vaccine (AZD1222 of Oxford/AstraZeneca, University of Oxford, UK) in two doses with an interval of 8–12 weeks, as well as 67 individuals (male/female: 26/41, median age: 47.0 years, range: 20–74) who received the Ad.26.COV2.S COVID-19 vaccine (Janssen Biotech, Inc, Janssen Pharmaceutical Company, Johnson & Johnson, New Jersey, NJ, USA) in one dose in accordance with the Greek Ministry of Health guidelines. Among them, 11 individuals (8.2%, all in the group of J&J/Janssen COVID-19 vaccine) exhibited a positive medical history of COVID-19 prior to vaccination. Anti-N IgG anti-SARS-CoV-2 antibodies were additionally detected in one participant at enrollment (this participant also received the J&J/Janssen COVID-19 vaccine); therefore, the total number of participants with a previous history of COVID-19 was 12 (8.9%). Consequently, the total number of enrolled participants in the study was 476 (male/female: 216/260, median age: 55.0 years, range: 20–105) derived from both the community and several health care facilities to collect real-world data from a broad range of age groups. Appendix A presents an overview of the enrolled individuals’ demographic data, medical history, and adverse effects after vaccination.

Serum and whole blood samples were collected at: (a) day 21; (b) day 42; and (c) day 90 after the first dose. Particularly for the participants receiving the BNT162b2 mRNA vaccine, blood collection on day 21 was performed the day of the second dose and just before the vaccination. Anti-spike protein IgG and IgA levels were analyzed in all individuals until day 90 as IgA response was found to profoundly decrease thereafter [20]. For participants vaccinated with BNT162b2, samples were collected between January to May 2021, while for those vaccinated with the ChAdOx1 nCoV-19 and Ad.26.COV2.S COVID-19 vaccines, samples were collected between March to July 2021 and May to September 2021, respectively. Predominant variants of SARS-CoV-2 in Greece during the vaccination period were Alpha (B.1.1.7/UK lineage) and Delta (B.1.617.2) (manuscript in preparation). The study was designed to assess antibody responses after the initial doses of COVID-19 vaccination and terminated in March 2022. All participants, either infected or not, received booster vaccinations with BNT162b2 in accordance with guidelines of the National Vaccination Committee of Greece (the analysis of antibody responses to post-booster vaccinations are pending and will be findings of a future study).

All enrolled individuals provided informed consent. The study was conducted according to the principles of the Helsinki Declaration and was approved by the ethical committee of the Faculty of Medicine, University of Thessaly (No. 2116, 22 April 2020).

### 4.2. Humoral Serum Responses to Vaccination

Anti-spike (S1) anti-SARS-CoV-2 IgG and IgA antibodies were detected in serum with commercially available kits, as described. In particular, the presence of anti-SARS-CoV-2 anti-S IgG antibodies was determined using an ABBOTT SARS-CoV-2 IgG II assay, a chemiluminescent microparticle immunoassay (CMIA) with an ARCHITECT i2000SR immunoassay analyzer (Abbott, IL, USA). The measurement of anti-SARS-CoV-2 anti-S IgA antibodies was estimated using a SERION ELISA agile SARS-CoV-2 IgA ESR400A kit (Serion, Würzburg, Germany) according to the manufacturer instructions. Moreover, anti-N anti-SARS-CoV-2 IgG antibodies, determined as described by an ABBOTT SARS-CoV-2 IgG II assay [35], were also used as markers of SARS-CoV-2 infection before and after vaccination.

### 4.3. IgA Saliva Responses to Vaccination

IgA responses in the saliva of nine participants (male/female: 5/4, median age 42 years: range: 32–57)—of whom four carried the wild-type rs1883832 alleles, two were heterozygotes, and three were homozygotes—were tested 30–40 days after a third dose with the BNT162b2 mRNA vaccine. All these participants had initially received the BNT162b2 mRNA vaccine as well. More than 1 mL of saliva was collected from each participant and centrifuged for 5 min at 13.000 rpm. The supernatants were filtered twice, initially through a cell strainer with 50 μm mesh (Celltrics, Sysmex, Germany), and after through a plug syringe outlet into 0·46 μm filter (Pall Corporation, Hampshire, UK). Aliquots of 200–500 μL were then stored at −80 °C and thawed on ice for further analysis. The measurement of anti-SARS-CoV-2 anti-spike IgA antibodies was determined as described above.

### 4.4. Molecular Techniques

Genomic DNA was extracted from peripheral blood using the EXTRACTME GENOMIC DNA KIT (Blirt, Gdansk, Poland), according to manufacturer instructions. The detection of CD40-1C>T polymorphism (rs1883832) was performed by PCR amplification of a 217-bp fragment that encompasses the Kozak region of the *TNFSF5* gene, followed by restriction fragment length polymorphism (RFLP) analysis. Specifically, a total of 100–200 ng of DNA was amplified in a 30 μL reaction mixture using 62·5 μM of each deoxynucleoside triphosphate, 20pmol of each primer (forward: 5′-ATAGGTGGACCGCGATTGGT-3′, reverse: 5′- TCCCAACTCCCGTCTGGT-3′), 1.5 mM MgCl2, and 0.8 U DFS-Taq DNA polymerase (Bioron GmbH, Romerberg, Germany) in a buffer supplied by the manufacturer. The thermocycler conditions of PCR consisted of an initial denaturation at 95 °C for 2 min, followed by 32 cycles of amplification (denaturation at 95 °C for 30 s, annealing at 58 °C for 30 sec, and elongation at 72 °C for 30 s) and a final elongation at 72 °C for 10 min. Afterwards, the PCR products were subjected to digestion with the restriction enzyme NcoI (New England Biolabs, Ipswich, UK) at 37 °C overnight. The presence of undigested PCR products was indicative of homozygote samples (TT alleles), whereas the presence of the wild-type allele (C) resulted in the digestion of the PCR product to 127-bp and 90-bp fragments. For the confirmation of the PCR-RFLP results, randomly chosen wild-type, heterozygous and homozygous PCR products were purified via a PCR purification system (Qiagen, Crawley, UK) and were directly sequenced using an ABI Prism 310 genetic analyzer (Applied Biosystems, Foster City, CA, USA) and a BigDye Terminator DNA sequencing kit (Applied Biosystems).

### 4.5. Immunophenotyping Studies

Peripheral blood samples were collected from 19 individuals (male/female: 8/11, median age: 51 years, range: 31–67) according to their rs1883832 polymorphism status, including eight with wild-type alleles, six heterozygotes, and five homozygotes. Immunophenotyping was performed by flow cytometry on a Coulter FC-500 instrument (Epics XL-MCL, 4 color analysis, Beckman-Coulter/BC, Hialeah, FL, USA) using a multi-staining protocol and commercially available reagents. Mouse anti-human IgG monoclonal antibodies were used to detect molecules that reacted with CD19 (clone J3-119), CD40 (clone MAB89), and CD20 (clone B9E9). All antibodies were purchased by BC and were conjugated with the appropriate fluorochrome (FITC, PE, PE-Cy5). The percentage of fluorescent cells and the MFI were determined and corrected for background fluorescence using FITC, PE, and PE-Cy5-labelled control IgG antibodies. The samples from each group were collected and analyzed on the same day to minimize day-to-day variation, especially for the MFI measurement.

### 4.6. Statistical Analysis

Categorical variables were described using frequency and relative frequency. Continuous variables were described with the use of median and interquartile range (IQR). Analysis of continuous variables was conducted using the Mann–Whitney U test and correlations were made using Spearman’s rank correlation coefficient. Data were checked for deviation from normal distribution using the Shapiro–Wilk normality test. Multivariable analysis was performed using multiple regression and Cox regression techniques. For all analyses, a 5% significance level was set. Analysis was carried out with SPSS (version 25.0) and GraphPad Prism (version 9.2.0) software.

## Figures and Tables

**Figure 1 ijms-23-14056-f001:**
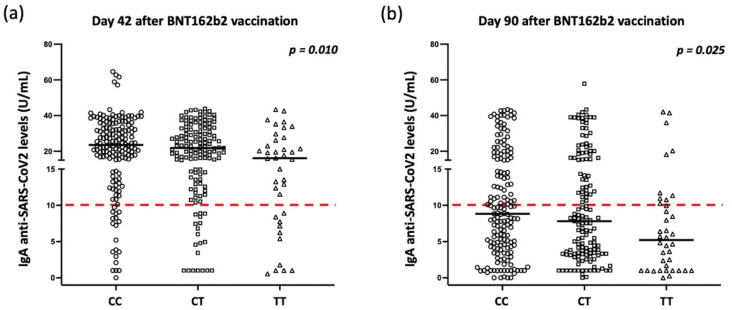
Serum IgA levels according to the status of CD40 rs1883832 polymorphism for participants vaccinated with the BNT162b2 mRNA vaccine (**a**) on day 42 after the first vaccination and (**b**) on day 90 after the first vaccination. Black lines indicate the median values, and red dotted lines represent the cut-off of positive anti-SARS-CoV-2 IgA (10 U/mL) antibodies. The statistical analysis was performed by Kruskal–Wallis H test.

**Figure 2 ijms-23-14056-f002:**
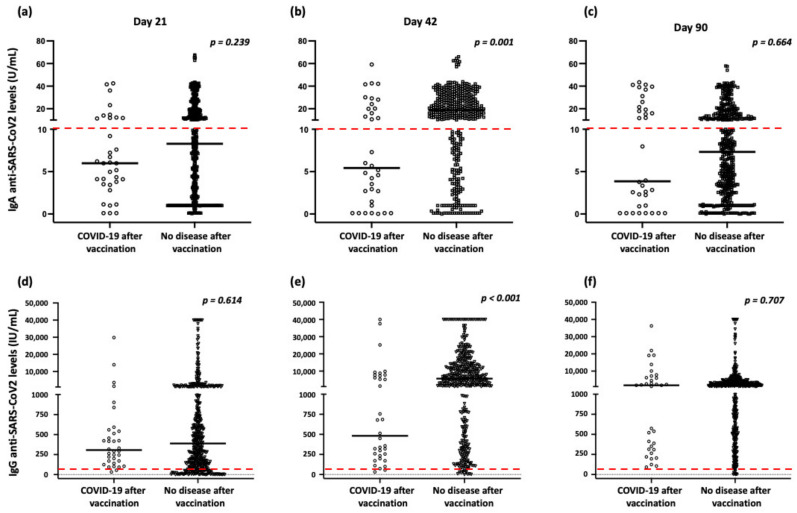
Serum IgA and IgG levels according to development of COVID-19 after vaccination: (**a**) IgA levels on day 21 after vaccination; (**b**) IgA levels on day 42 after vaccination; (**c**) IgA levels on day 90 after vaccination; (**d**) IgG levels on day 21 after vaccination; (**e**) IgG levels on day 42 after vaccination; (**f**) IgG levels on day 90 after vaccination. Black lines indicate the median values, and red dotted lines represent the cut-off of positive anti-SARS-CoV-2 IgA (10 U/mL) and anti-IgG (50 AU/mL) antibodies. The rectangle scheme highlights the statistical significance of IgA and IgG levels on day 42. The statistical analyses were performed by Mann–Whitney U test.

**Table 1 ijms-23-14056-t001:** Association of rs1883832 polymorphism with IgA responses (U/mL) after COVID-19 vaccination.

	Participants with Genotype			
	CC (wt)	CT (het)	TT (hom)	*p* *
BNT162b2 vaccine (Pfizer-BioNTech) (No 342)				
No (%)	161 (47.1)	143 (41.8)	38 (11.1)	
Day 21 (median, range)	9.6, 0.0–64.9	7.8, 0.4–43.5	6.2, 1.0–43.4	0.159
Day 42 (median, range)	23.5, 0.0–64.6	21.9, 1.0–43.9	16.1, 0.6–43.4	0.010
Day 90 (median, range)	8.8, 0.0–43.4	7.8, 0.0–57.9	5.2, 0.0–42.1	0.028
Adenovirus-based vector vaccines (No 134)				
No (%)	55 (41.1)	65 (48.5)	14 (10.4)	
Day 21 (median, range)	6.7, 0.1–67.6	6.8, 0.1–65.4	7.4, 0.1–41.6	0.665
Day 42 (median, range)	5.3, 0.0–66.0	2.86, 0.1–62.5	3.0, 0.1–41.6	0.078
Day 90 (median, range)	4.8, 0.1–57.5	2.69, 0.1–41.1	7.6, 0.1–41.3	0.296
Total (No 476)				
No (%)	216 (45.4)	208 (43.7)	52 (10.9)	
Day 21 (median, range)	9.0, 0.0–67.6	7.3, 0.1–65.4	6.4, 0.1–43.4	0.125
Day 42 (median, range)	20.6, 0.0–66.0	16.2, 0.1–62.5	13.1, 0.1–43.4	0.005
Day 90 (median, range)	8.2, 0.0–57.5	6.6, 0.0–57.9	5.6, 0.0–42.1	0.053

Abbreviations: wt, wild type; het, heterozygous; hom, homozygous. (*) The statistical analysis was performed by Kruskal–Wallis H test.

**Table 2 ijms-23-14056-t002:** Association of rs1883832 polymorphism with IgG responses (AU/mL) after COVID-19 vaccination.

	Participants with Genotype			
	CC (wt)	CT (het)	TT (hom)	*p* *
BNT162b2 vaccine (Pfizer-BioNTech) (No 342)				
No (%)	161 (47.1)	143 (41.8)	38 (11.1)	
Day 21 (median, range)	438.9, 0.0–40,000.0	467.9, 0.0–40,000.0	293.1, 0.5–40,000.0	0.563
Day 42 (median, range)	9107.3, 0.2–40,000.0	9314.6, 1.6–40,000.0	7024.3, 8.2–40,000.0	0.571
Day 90 (median, range)	2097.2, 3.2–40,000.0	2303.5, 0.0–40,000.0	1757.4, 6.4–40,000.0	0.677
Adenovirus-based vector vaccines (No 134)				
No (%)	55 (41.1)	65 (48.5)	14 (10.4)	
Day 21 (median, range)	336.0, 7.1–33,994.2	285.0, 7.8–13,966.7	292.9, 135.7–29,806.1	0.508
Day 42 (median, range)	364.1, 29.6–26,246.5	298.2, 22.4–11,215.5	315.0, 85.7–9827.3	0.244
Day 90 (median, range)	839.7, 93.5–40,000.0	713.6, 43.2–36,241.2	677.2, 276.2–19,093.0	0.371
Total (No 476)				
No (%)	216 (45.4)	208 (43.7)	52 (10.9)	
Day 21 (median, range)	414.5, 0.0–40,000.0	362.5, 0.0–40,000.0	293.1, 0.5–40,000.0	0.559
Day 42 (median, range)	6173.1, 0.2–40,000.0	4579.7, 1.6–40,000.0	4857.8, 8.2–40,000.0	0.223
Day 90 (median, range)	1783.1, 3.2–40,000.0	1628.9, 0.0–40,000.0	1601.7, 6.4–40,000.0	0.531

Abbreviations: wt, wild type; het, heterozygous; hom, homozygous. (*) The statistical analysis was performed by Kruskal–Wallis H test.

**Table 3 ijms-23-14056-t003:** Multivariable analysis exploring the risk factors affecting the anti-SARS-CoV-2 IgA levels at days 42 and 90 after vaccination.

Parameter	Coefficient, 95% CI	*p*
Day 42 after vaccination		
Sex (female/male)	−0.54, −2.69–1.60	0.617
Age	−0.12, −0.18–−0.07	<0.001
History of COVID-19 before vaccination	18.07, 14.70–21.43	<0.001
History of COVID-19 after vaccination	−0.68, −4.87–3.52	0.752
rs1883832 polymorphism (het and hom vs. wt)	−4.00, −6.06–−1.94	<0.001
Vaccination type (BNT162b2 vs. adenovirus-based vector vaccines)	6.93, 5.71–8.15	<0.001
Day 90 after vaccination		
Sex (female/male)	−1.39, −3.21–0.43	0.135
Age	−0.02, −0.07–0.03	0.443
History of COVID-19 before vaccination	21.78, 18.92–24.64	<0.001
History of COVID-19 after vaccination	4.59, 1.02–8.16	0.012
rs1883832 polymorphism (het and hom vs. wt)	−2.43, −4.18–−0.88	0.007
Vaccination type (BNT162b2 vs. adenovirus-based vector vaccines)	1.40, 0.36–2.44	0.008

Abbreviations: wt, wild type; het, heterozygous; hom, homozygous.

## Data Availability

The data that support the findings of this study are available from the corresponding authors upon request.

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
