# Peer review of "The rs1883832 Polymorphism (CD40-1C>T) Affects the Intensity of IgA Responses after BNT162b2 Vaccination"

_ijms, 2022, doi:10.3390/ijms232214056_

Round 1

Reviewer 1 Report

The paper entitled “The rs1883832 polymorphism (CD40-1C>T) affects the intensity of IgA responses after BNT162b2 vaccination” describes how the response to COVID-19 vaccination is affected in individuals carrying the rs1883832 polymorphism. This study enrolled 476 participants vaccinated in several vaccine regiments, of those 52 participants are homozygous for this polymorphism. The allele frequencies of this polymorphism are described as Hardy-Weinberg equilibrium. There is no description of whether this equilibrium is found in other genome wide studies, nor is it representative of the Greek population. Furthermore, if carrying 2 alleles of the rs1883832 polymorphism do lead to a significant change in response to vaccination, one would expect the Hardy-Weinberg equilibrium to be affected. This should be addressed.  

There is no clear description of who are the participants of this study. Is it general population, what are the inclusion/exclusion criteria. Furthermore, it needs to be clearly described how the study was conducted: how many visits each participant had, what was collected in each visit. Also, since the pandemic had multiple infection waves it is valuable to know when the recruitment happened and which variant was the dominant one at the time, is the infection post 2 vaccine doses was mostly during the omicron wave? Finally, there is no mention of recipient of the boost shot, even if the study was terminated before the booster became commonly used it needs to be mentioned.

Table 1:is not clear to whether COVID-19 infection as well as adverse events are reported after 1 or 2 doses. It is not clear why the authors decided to present this extensive medical history in table 1. The history is not mentioned elsewhere in the paper and should be moved to the supplementary.

The authors describe the antibody responses post first vaccine dose, in table 2. The recommended gap between 2 doses is 3 weeks, so the first timepoint (at day 21) should have been after the second dose, is that the case here? This should be clearly stated. The reduction in anti IgA levels is significantly lower only the 42-day timepoint. There is no explanation to why this is important, or what is the relevance of that if after 90 days there is no significance difference in the decay between the groups.

·         There is a problem in the headings of both tables 2 and 3, C/C is presented twice, both for wt and hom

·         table 4: a problem with headings Day 42 appears twice (I assume the second one should have been day 90)

Figure 1 shows the same data presented in table 2, not sure it needs to be presented twice in the main text, it can be moved to the supplementary.

Table 4 shows that age is a risk factor to lower IgA levels. Age has been reported by many others to be associated with lower response to vaccine, this needs to be mentioned in the discussion. I expected to see the multivariant analysis for IgG levels as well.

Figure 2 is showing the IgA levels in the saliva, which is the main site for testing IgA levels. The figure does not have any statistical analysis. It should be added, and stated if significant or not (for all panels). Panel B, shows the MFI of CD40, claiming to be lower in the participants carrying 2 alleles of rs1883832 polymorphism. This is not clear from the figure. It is also not clear which samples the histogram represents, and the axis are not labeled, it much be added. overall it is not convincing that this polymorphism leads to a significant change in the CD40 expression levels.

Figure 3 describes the IgA and IgG levels in participants being infected or not with COVID-19. As mentioned in the text there is not association with the rs1883832 polymorphism and infection. Therefore, the conclusion that IgA levels are important for protection needs to be further assessed, and extensively discussed in the discussion part.

There is in no mention of the following publication, which discusses the relevance of IgA responses to vaccines and infection: Sheikh-Mohamed, S., Isho, B., Chao, G.Y.C. et al. Systemic and mucosal IgA responses are variably induced in response to SARS-CoV-2 mRNA vaccination and are associated with protection against subsequent infection. Mucosal Immunol 15, 799–808 (2022). https://doi.org/10.1038/s41385-022-00511-0

Overall, this work brings up an interesting observation linking a genetic polymorphism with the ability to vaccinate. It needs to be better presented in all aspects: introduction, study design and the conclusion drawn from it (should be more modest). Moreover, it needs to cover more of the existing literature and better link the work presented here to what is already published.

Author Response

Please see the attachment, whereas ALL the comments and recommendations of both reviewers have appropriately been addressed. Considering that some of the them are common, we upload our responses to both reviewers

Reviewer 2 Report

The rs1883832 polymorphism (CD40-1C>T) affects the intensity of IgA responses after BNT162b2 vaccination by Matthaios Speletas and co-authors is interesting on some respects but is difficult to fully evaluate given some sloppiness in the data presentation and the considerable liberties taken by the authors in data interpretation based on very small numbers of subjects.

Major Comments

Pg 7 Ln 179-183

This is the first time in the manuscript that booster immunization is mentioned. There is no indication of who is being boosted, what their primary series was, their demographics, etc. The vaccine used to boost is not even mentioned! Furthermore … it appears that their finding of reduced CD40 expression on peripheral blood B cells is limited to the small number of ‘boosted’ subjects (N=8 CC, N=6 C/T and N=4 TT)

Pg 7 Ln 199

It would appear (from the number of data points) that the analysis of susceptibility to infection includes all vaccines rather than just those in whom the purported effect of the T allele on IgA was demonstrated (ie: mRNA vaccine recipients). This doesn’t make a great deal of sense to me. There also appears to be a striking effect of D42 IgG titers on susceptibility to infection (something widely understood as a decent correlate of protection for all vaccines) without reference to the allele under consideration. Finally … there is no description of when the apparent breakthrough infections occurred in these subjects (ie: between D42 and D90, after D90, etc) or the epidemiology of SARS-COV-2 variants during the study (ie: what variants were circulating at the time of the study or (better) what variants caused the breakthrough case?)

Page 8 Ln 212

The authors state that ‘We clearly demonstrate a significant association of the rs1883832 polymorphism of 213 TNFRSF5 gene to anti-spike IgA antibody responses after vaccination with the SARS-214 CoV-2 S1 protein.’ which is a broad over-statement. They may have shown this effect in recipients of one of the mRNA vaccines at two time-points after vaccination (D42 and D90). They then grossly overstate their finding of (possible) differences in salivary IgA and CD40 expression on B cells which are based on very limited data and does not appear to have reached statistical significance.

Table 1: This Table is inadequately labeled so the reader has to guess (for example) what is the N and what is the % for some parameters.

Figure 1: Since the authors demonstrate that very little IgA is induced by the adenovirus-vectored vaccine, it is not clear to me why the inclusion of D42 samples from these subjects is informative (Figure 3a). On the contrary, the apparently ‘greater significance’ of the impact of the T/T homozygous state (p<0.001) is likely spurious sine it is driven by inclusion of the non-responding adenovirus subjects (ie: comparing Figure 3a and 3b).

Table 4: The lower part of Table 4 appears to be mislabeled since both sections are labeled D42 after vaccination rather than D42 and D90 (as per the Table title).

Figure 2: It is not clear from the Figure legend what the histograms in part b actually represent? They are inadequately labeled.

Figure 3: The ‘red dotted line’ indicating the detection limit for IgA is missing in parts a, b and c. The most problematic aspect of Figure 3 is (again) the inclusion of all subjects rather than only the mRNA vaccinated subjects in whom the purported effect of the T allele was seen. Since we know that the adenovirus-vectored vaccines induced lower levels of IgG overall and very little IgA, their presence in this Figure is potentially quite misleading.

Author Response

Please see the attachment, where ALL the comments and recommendations of both reviewers have been addressed. Considering that some of them are common, we upload our responses to both reviewers

Round 2

Reviewer 1 Report

1)       fig 2 consider removing the rectangle

2)       figS2a need to add red dotted line for the saliva graph.

figS2b- the histograms lack labels on the X axis and the Y axis need to be better labeled in the quantification graph. In needs to say if the scale is logarithmic. Also, MFI are usually present in the thousands.  

Since there is no statistically significant difference between the groups and the stud material collection is finished one does not now is this is not significant due to smaller number of participants or simply since it is not significant. This is a wee point of this work and authors might consider to remove it altogether.

3)       Age being a risk factor, has been proposed by multiple groups, this is an example:

Liang W, Liang H, Ou L, et al. Development and validation of a clinical risk score to predict the occurrence of critical illness in hospitalized patients with COVID-19. JAMA Intern Med 2020;180:1081–1089.

Please review the literature and cite the important findings in this regard.

4)       Multivariant analysis for IgG is still missing.
